# Inferences about Fishing Impacts on the Reproductive Pattern of Torroto Grunt—*Genyatremus luteus* (BLOCH, 1790)—From the Amazon Continental Shelf

**Suelly Fernandes [1], Larissa Pinheiro [1,2,*] and Bianca Bentes [1,2]**

[1] Núcleo de Ecologia Aquática e Pesca da Amazônia, Universidade Federal do Pará-UFPA, Belém 66077-530, Brazil
[2] Programa de Pós-graduação em Ecologia Aquática e Pesca, UFPA, Belém 66077-530, Brazil
* Correspondence: larissap.biologia@gmail.com

**Abstract:** *Genyatremus luteus'* reproductive dynamics were investigated by examining the gonadal first maturation length ($L_{50}$) and reproductive period (gonadosomatic index, GSI, and relative condition factor, Kr) in relation to the environmental factor rainfall. Sampling was conducted monthly from April 2016 to April 2017, in five areas (A1 to A5) using gill nets and fixed traps. The sex ratio distribution of the 331 specimens captured (174 females, 140 males, and 17 with undetermined sex) did not differ between months. Females were larger (19.28 cm vs. 16.63 cm to males) and heavier (147.21 g vs. 92.14 g to males) and, for both sexes, the means of Kr and GSI indicated the probable spawning period of the species that occurred from June to September, after a decrease in the precipitation of the region. The value of $L_{50}$ was estimated as 15.13 cm for females, 14.29 cm for males, and 14.78 cm for both sexes. In the rainy season, the behavior of the species is associated with breeding in certain areas but in the dry season, it is related to the fattening and refuge of juveniles and adults. This research shows the first insight for fishing management about a species that requires different ecosystems to complete their life cycle.

**Keywords:** haemulidae; amazon rainfall; spawning period; $L_{50}$; sustainability development objective of United Nations-SDO#14

**Key Contribution:** This paper investigates the reproductive dynamic of *Genyatremus luteus*, pointing out that the completion of their life history needs different ecosystems. The research shows the impact of fishing on *Genyatremus luteus*'s reproduction and raises the challenges for fishery management.

## 1. Introduction

Fish throughout evolutionary history have developed differentiated reproductive strategies to adapt to environmental and biological changes in aquatic ecosystems [1]. These adaptations are associated with animal welfare (homeostasis) [2], which is fundamental in stabilizing vital body functions and reproductive processes [3]. Thus, the spawning season is directly associated with the guarantee of food supply to early life stages and a decrease in their natural mortality rates. This "environmental trigger" [4] is usually related to seasonal periods of high rainfall in the Amazon region, where fish species have synchronized their most critical physiological activities, such as the reproduction and consequent entry and exit of larvae, juveniles, and adults, to environmental processes caused by rainfall [5,6].

As a marine estuarine species, *Genyatremus luteus* (Figure 1) (or peixe pedra in Amazon estuaries) follows a metabolic pattern related to environmental conditions [7–10]. *G.luteus* is classified as oviparous, with pelagic eggs and larvae [11] that use the estuary for growth and spawning [10].

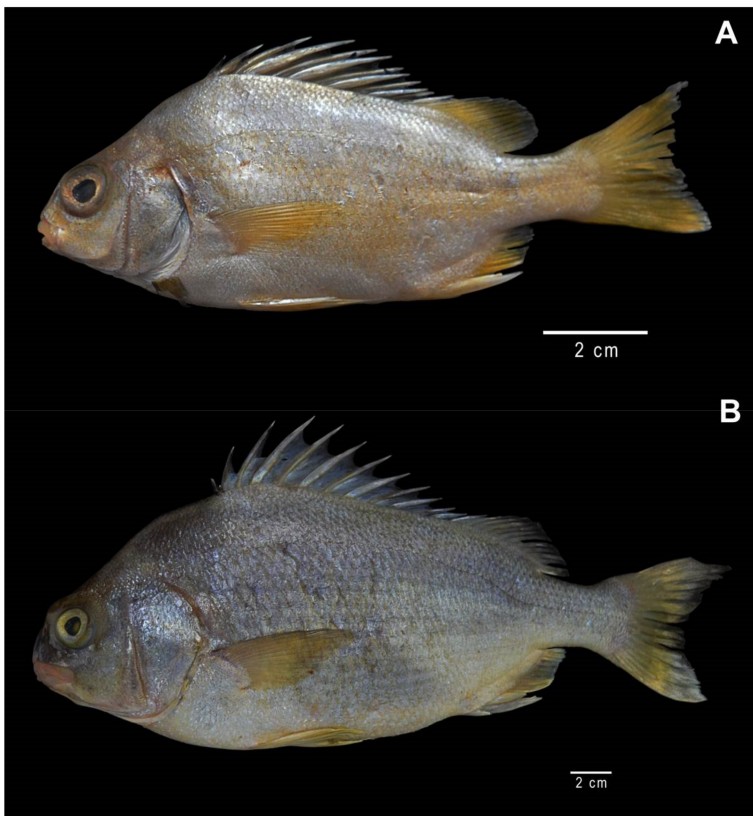

**Figure 1.** The specimen of *Genyatremus luteus* (BLOCH, 1790). (**A**): juvenile specimen; (**B**): adult specimen.

In the Amazon coastal estuarine areas, *G. luteus*, a commercially important species, is targeted by several fishing gears [9], which may compromise the ability of the species to complete its life cycle and self-renewal of the local population. Therefore, the objective of this study is to estimate the size of the first gonadal maturation ($L_{50}$) and to define the spawning season with respect to the environmental factor of rainfall, to understand the life cycle of the species, and to make inferences about the possible impacts of fishing activity.

## 2. Materials and Methods

Sampling was conducted monthly between April 2016 and April 2017 in five areas around one amazon estuary (Ajuruteua peninsula) (Figure 2). The captures were made during the waning lunar phase, in the ebb or flood tide, using gill nets (40, 45, 50, 60, and 70 mm between opposing knots) and traps [12] traditionally used by the commercial fishery in the region.

After capturing the specimens, the material was frozen and sent to the fisheries biology laboratory of the UFPA-Bragança campus, where its total length (TL in centimeters) and weight (in g) were recorded. After ventral incision, gonads were removed, weighted for the GSI calculation, and the developmental stage was determined macroscopically according to the methodology of Brown-Peterson et al. [13] as follows: immature (IM), development (D), spawning capacity (SC), regression (RG), and regeneration (RE). The monthly sex ratio was tested by using chi-square with a 5% error.

The average monthly rainfall of the last 20 years was obtained using data provided by the National Institute of Meteorology (INMET) from the Tracuateua station to test the seasonal trend recorded by Moraes et al. (2005) [14]. The rainy period (CP) occurs from January to July and the dry period (DP) occurs from August to December.

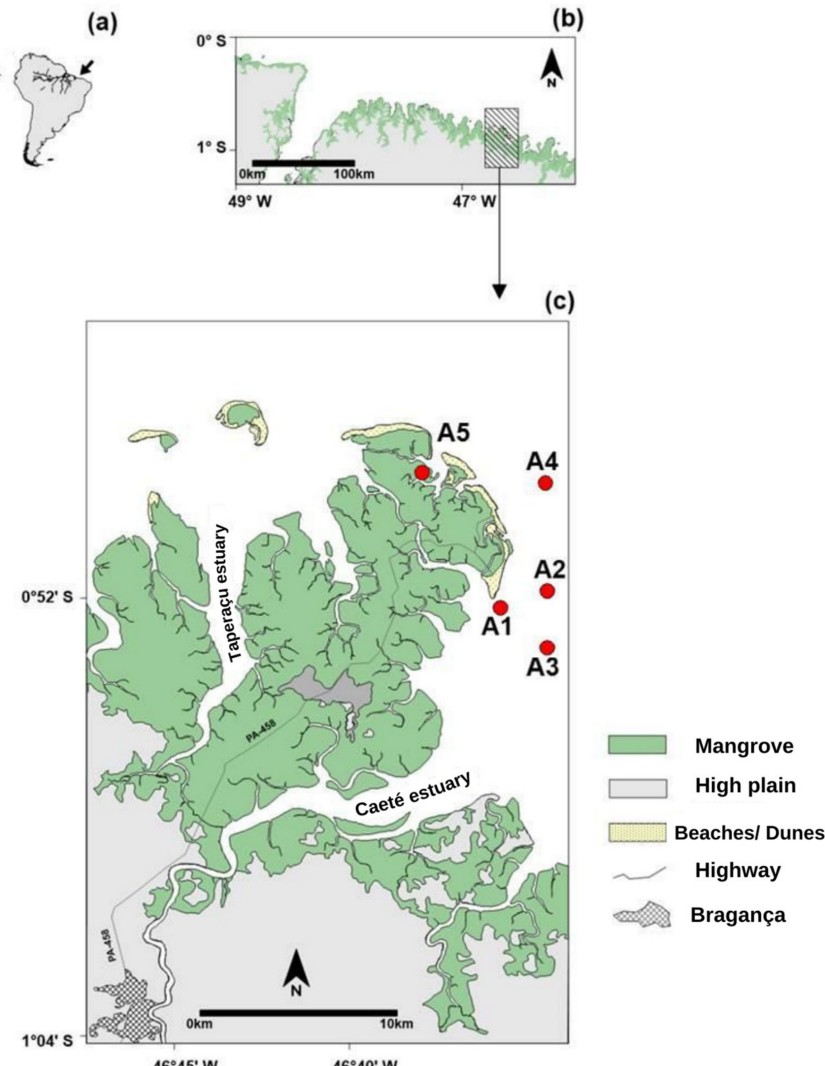

**Figure 2.** Geographic location area of study in the Brazil (**a**). Geographic area of study in the Amazon Continental Shelf (**b**). Geographic location of the Amazon estuary (Ajuruteua peninsula, Bragança, PA) (**c**) with the sampling areas (A1, A2, A3, A4, and A5) of *Genyatremus luteus*, carried out during the period from April/16 to April/17.

Total length (cm) and weight were tested between sexes using the *t* test ($\alpha = 0.05$). The mean length at first sexual maturity ($L_{50}$) was estimated for males, females, and sexes combined using the logistic regression of King (1995) [15]: $p = 1/1 + \exp(-r(L - L_{50})$, where $p$ is the proportion of adults; L: the maximum length obtained; $L_{50}$: the length corresponding to the proportion of 0.5 (50%) of the specimens in the analyzed conditions; and r: the angle of the curve, grouping the total lengths in intervals of 1 cm.

The spawning period was estimated using the relative condition factor (Kr) [16] and the gonadosomatic index (GSI) [17]) for females, males, and sexes combined. GSI and Kr are useful to predict the average weight at a given length or age group and then convert the length and data into weights to provide a major of biomass [18], because stock assessment models and management for fisheries require information about body weight for the estimation and regulation of catches [19] and weight-at-age [20]. Kr was obtained using the equation: Kr = Wobs/Wexp, where Wobs = the observed weight (g) of each specimen and Wexp = the weight (g) determined using non-linear TL-W regression. Weights were determined via the non-linear curve of the relationship between the total length and observed weight ($y = a*x^b$). To calculate the GSI the equation GSI = Wg/Wc × 100 was used, where: Wg is the gonad weight (g) and Wb is the body weight (Wb = total weight, Wg).

To verify the seasonal period and the geographical location(s) of the increase or decrease in the reproductive intensity of the species, we performed the Canonical Redundancy Analysis (RDA) using the CANOCO 4.5 software (Software for Canonical Community Ordination). In this method, 9999 permutations were performed (Monte Carlo Method) to test the significance of the variables (GSI and Kr) that contributed to the variability of the data, and only the variables statistically significant via the analysis (ANOVA) ($p < 0.05$) were inserted in the model. For the independent variables (seasonal periods, maturity stages, and areas) separate matrices of presence and absence were built, where each row represented an individual, and these were related to a second matrix ('treatment'), in which the dependent variables were listed one by one.

## 3. Results

A total of 331 specimens of *G. luteus* were captured (Table 1). Samples include 174 females, 140 males, and 17 specimens of an undetermined sex. The sizes of the females ranged from 10.40 to 35.40 cm and the total weight from 22 to 771 g, while males ranged from 9.50 to 24.60 cm with the weight ranging from 16 to 252 g.

**Table 1.** Variation in total length (cm) and weight (g) of *Genyatremus luteus* specimens collected from April/16 to April/17 on the Ajuruteua peninsula, Bragança, Pará. Min: minimum, max: maximum, SD: standard deviation, N: number of specimens, F: female, M: male, UND: undetermined.

| Month/Year | Sex | N | Total Length (cm) | | Weight (g) | |
|---|---|---|---|---|---|---|
| | | | Min–Max | Average ± SD | Min–Max | Average ± SD |
| Apr/16 | F | 6 | 18.20–35.40 | 23.38 ± 5.66 | 114.00–771.00 | 277.67 ± 224.40 |
| | M | 4 | 13.90–22.90 | 17.75 ± 3.70 | 47.00–198.00 | 112.75 ± 64.01 |
| May/16 | F | 22 | 17.60–31.90 | 23.23 ± 3.21 | 88.00–549.00 | 235.73 ± 100.53 |
| | M | 14 | 14.80–23.20 | 19.16 ± 1.90 | 74.00–233.00 | 138.43 ± 43.84 |
| Jun/16 | F | 13 | 10.40–24.60 | 20.78 ± 4.20 | 22.00–302.00 | 194.46 ± 79.08 |
| | M | 5 | 9.50–21.90 | 17.18 ± 4.39 | 16.00–188.00 | 109.80 ± 65.36 |
| | UND | 1 | 11.2 | - | 23 | - |
| Jul/16 | F | 15 | 16.50–25.70 | 20.33 ± 2.31 | 78.00–315.00 | 159.60 ± 62.14 |
| | M | 11 | 14.50–20.10 | 17.28 ± 1.86 | 51.00–127.00 | 91.73 ± 24.51 |
| | UND | 1 | 13.6 | - | 46 | - |
| Aug/16 | F | 17 | 11.80–27.40 | 18.62 ± 4.29 | 28.00–371.00 | 135.71 ± 93.58 |
| | M | 22 | 10.80–20.10 | 15.49 ± 2.90 | 22.00–147.00 | 72.95 ± 39.77 |
| | UND | 1 | 10.1 | - | 19 | - |
| Sep/16 | F | 30 | 11.60–22.50 | 16.55 ± 2.73 | 26.00–206.00 | 84.40 ± 41.81 |
| | M | 30 | 11.10–21.20 | 16.00 ± 3.21 | 25.00–180.00 | 81.73 ± 47.59 |
| | UND | 6 | 10.40–18.60 | 13.27 ± 3.11 | 18.00–114.00 | 43.50 ± 33.95 |
| Oct/16 | F | 10 | 13.20–25.80 | 20.15 ± 3.97 | 41.00–289.00 | 152.20 ± 79.11 |
| | M | 17 | 11.30–22.60 | 16.26 ± 4.44 | 25.00–190.00 | 91.59 ± 65.32 |
| | UND | 6 | 10.10–12.90 | 11.44 ± 1.00 | 20.00–45.00 | 31.40 ± 9.26 |
| Nov/16 | F | - | - | - | - | - |
| | M | - | - | - | - | - |
| Dec/16 | F | 14 | 12.60–23.70 | 16.55 ± 3.50 | 36.00–211.00 | 90.50 ± 59.78 |
| | M | 17 | 12.00–24.60 | 17.69 ± 3.86 | 32.00–252.00 | 109.53 ± 69.12 |
| | UND | 1 | 16.2 | - | 65 | - |
| Jan/17 | F | 17 | 13.60–28.60 | 21.55 ± 3.64 | 42.00–385.00 | 190.29 ± 86.59 |
| | M | 8 | 11.50–21.40 | 16.28 ± 3.37 | 28.00–151.00 | 85.38 ± 40.27 |
| Feb/17 | F | - | - | - | - | - |
| | M | - | - | - | - | - |

**Table 1.** *Cont.*

| Month/Year | Sex | N | Total Length (cm) | | Weight (g) | |
|---|---|---|---|---|---|---|
| | | | Min–Max | Average ± SD | Min–Max | Average ± SD |
| Mar/17 | F | 13 | 12.40–19.10 | 15.37 ± 2.16 | 32.00–120.00 | 68.92 ± 28.11 |
| | M | 5 | 12.50–17.90 | 15.54 ± 1.92 | 35.00–98.00 | 68.00 ± 25.18 |
| | UND | 2 | 13.40–14.00 | 13.70 ± 0.30 | 45.00–49.00 | 47.00 ± 2.00 |
| Apr/17 | F | 17 | 14.10–25.40 | 18.56 ± 3.27 | 47.00–305.00 | 122.41 ± 69.03 |
| | M | 7 | 13.00–17.80 | 15.37 ± 1.67 | 37.00–106.00 | 65.29 ± 22.79 |
| Total | F | 174 | 10.40–35.40 | 19.28 ± 4.28 | 22.00–771.00 | 147.21 ± 101.50 |
| | M | 140 | 9.50–24.60 | 16.63 ± 3.41 | 16.00–252.00 | 92.14 ± 53.82 |
| | UND | 17 | 10.10–18.60 | 12.66 ± 2.39 | 18.00–114.00 | 39.12 ± 23.35 |
| Total individuals | | 331 | 9.50–35.40 | 17.82 ± 4.23 | 16.00–771.00 | 118.37 ± 87.86 |

The highest individual mean sizes and weights occurred in April/16 (female: 23.38 ± 5.66 cm and 277.67 ± 224.40 g) and the lowest in October/16 (indeterminate: 11.44 ± 1.00 cm and 31.40 ± 9.26 g). On average, females were larger than males in length (females: 19.28 ± 4.28 cm; males: 16.63 ± 3.41; t = 0.054; $p < 0.05$) and wet weight (females: 147.21 ± 101.50 g; males: 92.14 ± 53.82; t = 0.0102; $p < 0.05$).

The monthly relative frequency of females and males did not differ statistically ($p > 0.05$). No specimen captures were recorded in November and February.

The specimens captured in this study were largely immature (92 individuals, the highest number). Other gonadal stages observed included 85 in regression, 78 with spawning ability, 55 in development, 15 in regeneration, and 6 indeterminate. The immature individuals were distributed in the size range of 9.50 to 20.10 cm (Figure 3). The gonadal maturation stage where individuals were in a spawning capacity started in the size range of 12.15–14.80 cm (Figure 3).

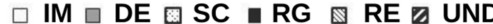

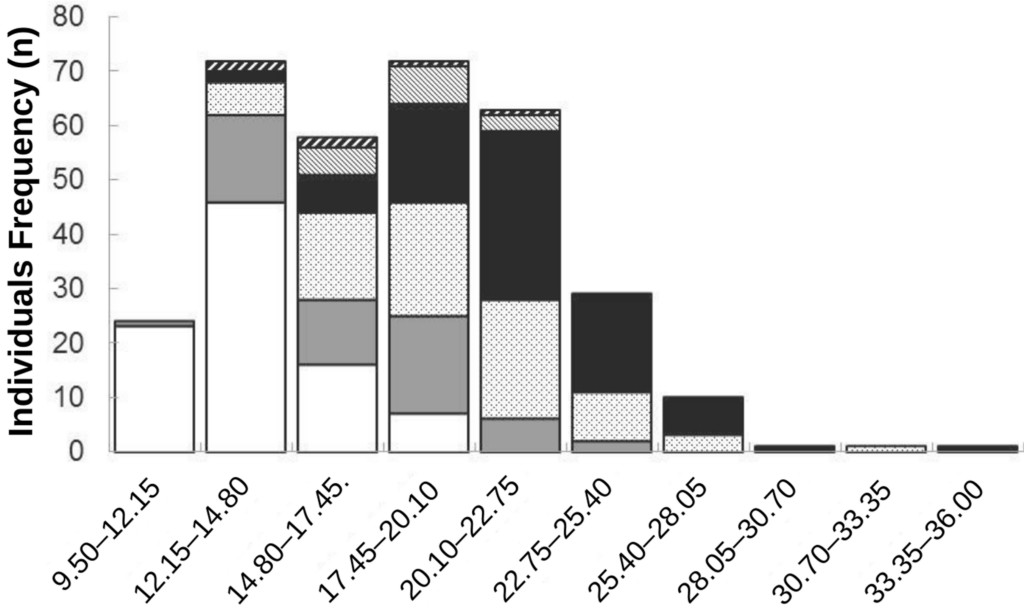

**Figure 3.** Frequency distribution of specimens by gonad stage was IM: immature, DE: development, SC: spawning-capable, RG: regression, RE: regeneration, UND: undetermined. Total length (TL, cm) of *Genyatremus luteus*, captured from April/16 to April/17, Ajuruteua peninsula, Bragança, PA, Brazil. Totaling 331 specimens.

The number of regressing females (n= 70) was higher than the other maturity stages (immature, n = 35; developing, n = 32; spawning-capable, n = 21; regenerating, n = 14; and indeterminate, n = 2). The spawning ability stage for males was more numerous, with 57 individuals. The other specimens comprise 44 immatures, 23 developing, 15 regressing, and only one (1) regenerating individuals. May had the highest predominance of gonads in the spawning-capable stage for females, and for males it was in September and May.

Females in the spawning-capable stage were observed between April–September, December, and March (Figure 4A). Spawning-capable males were observed in all months of this study (Figure 3B). No immature males were observed in March (Figure 4B). No immature females were observed in April or July (Figure 4A).

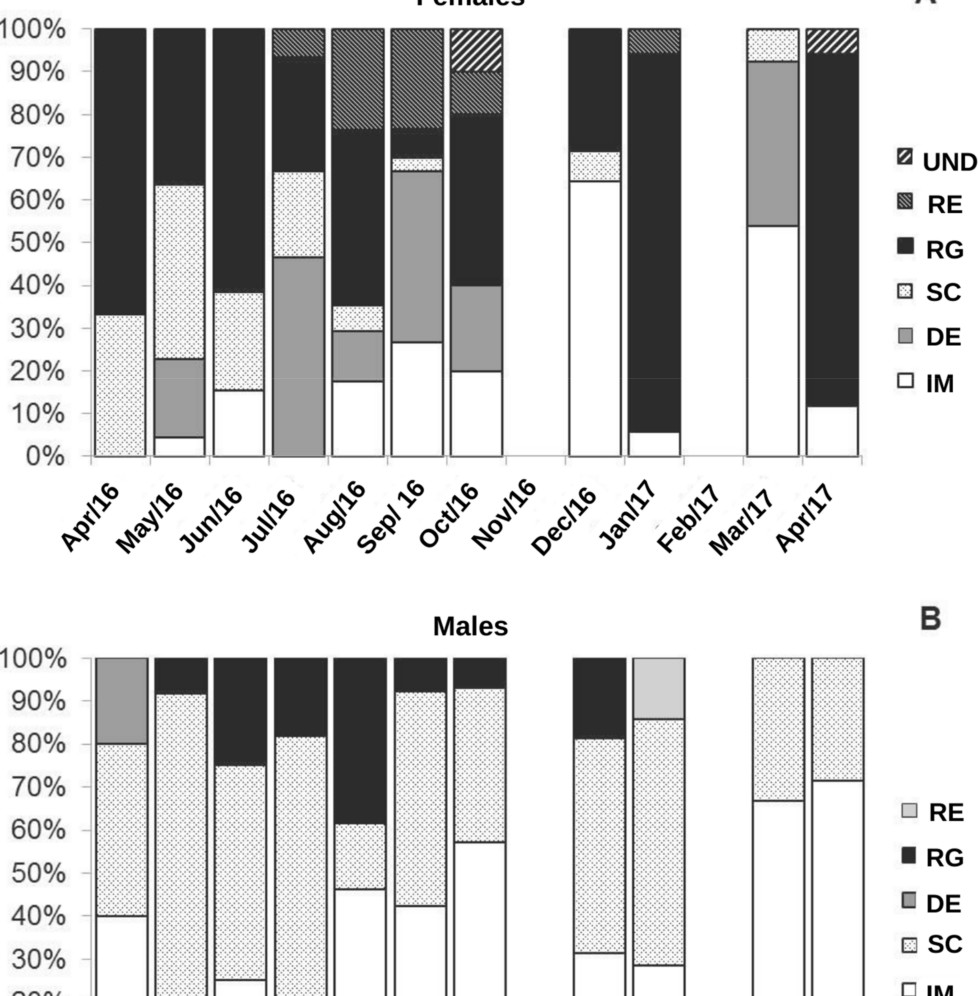

**Figure 4.** Relative percentage of gonadal maturation stages of females and males of *Genyatremus luteus* collected from April/16 to April/17 at Ajuruteua peninsula, Bragança, PA, Brazil. (**A**)—females, (**B**)—males, IM: immature, DE: development, SC: spawning-capable, RG: regressing, RE: regeneration, UND: undetermined.

For both sexes, the results of Kr and mean GSI were equivalent, indicating June–September as the probable spawning period of the species, with a greater reproductive

intensity in June. The mean GSI for females was higher in April and May and for males in January (Figure 5A). The mean Kr was higher in June for females and in May for males (Figure 5A). After the decrease in precipitation, the highest values of mean GSI for females and mean Kr for both sexes were observed. The decline in GSI was simultaneous to the decrease in precipitation until October for both sexes, while for Kr, this decline is visualized only for females. Males have different behaviors, where the decline in Kr occurs until July (Figure 5B).

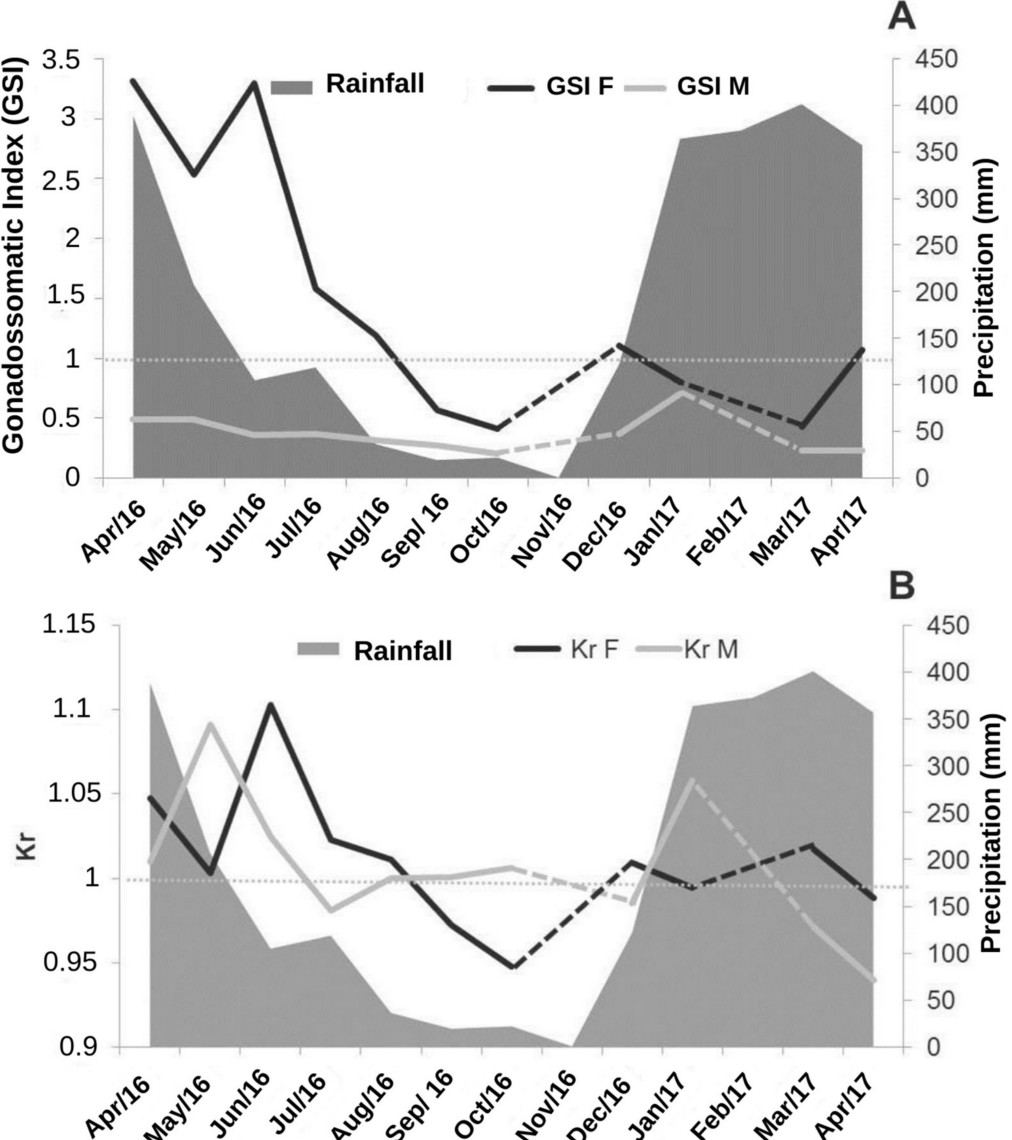

**Figure 5.** Average of gonadosomatic index by month (GSI-**A**) and relative factor condition (Kr-**B**) of *Genyatremus luteus* females and males in an Amazon estuary, Brazil, considering the precipitation level. GSI F: females' gonadosomatic index; GSI M: males' gonadosomatic index; Kr F: females' factor condition; Kr M: males' factor condition. Dashed lines gray and black indicate no specimens collected, even with the same sample effort. Dashed point line represents Kr = 1.

Estimated $L_{50}$ values were 15.13 cm for females, 14.29 cm for males, and 14.78 cm for sex combined (Figure 6).

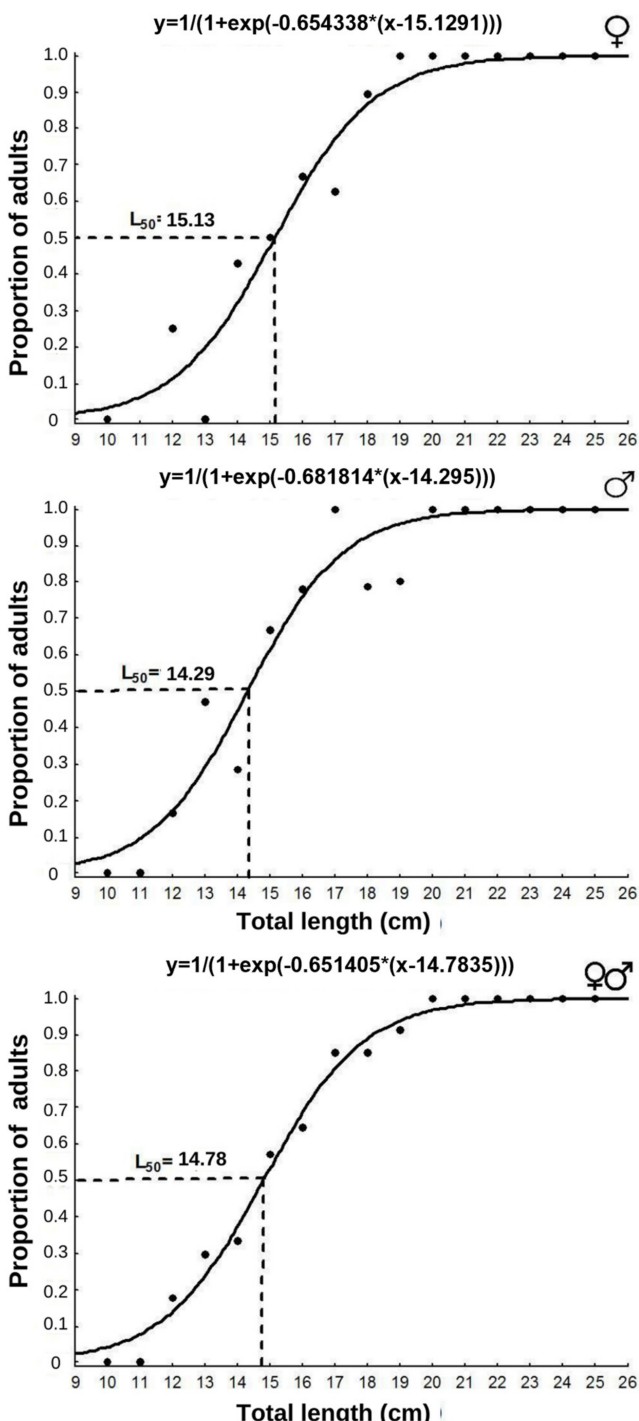

**Figure 6.** Size of first gonadal maturation of *Genyatremus luteus* captured from April/16 to April/17, Ajuruteua peninsula, Bragança, PA, Brazil.

The variable-correlated condition factor (Kr) and gonadosomatic index (GSI) were statistically significant (F = 3.158; $p < 0.01$ and F = 13.048; $p < 0.01$, respectively) between the months (percentage of cumulative variance of axis 1 = 82.10%). The GSI correlated positively to the rainy season for females with spawning capacity and in regressing females, in areas A1 and A2. GSI is negatively correlated with the dry period, males, or immature, developing, regenerating, and undetermined individuals, in areas A3 and A5 (Figure 7). The variable Kr correlated positively to the rainy period and immature, regressing, and

undetermined individuals, in areas A3 and A5, and negatively to the dry period for the individuals in regeneration at sites A2 and A4 (Figure 7).

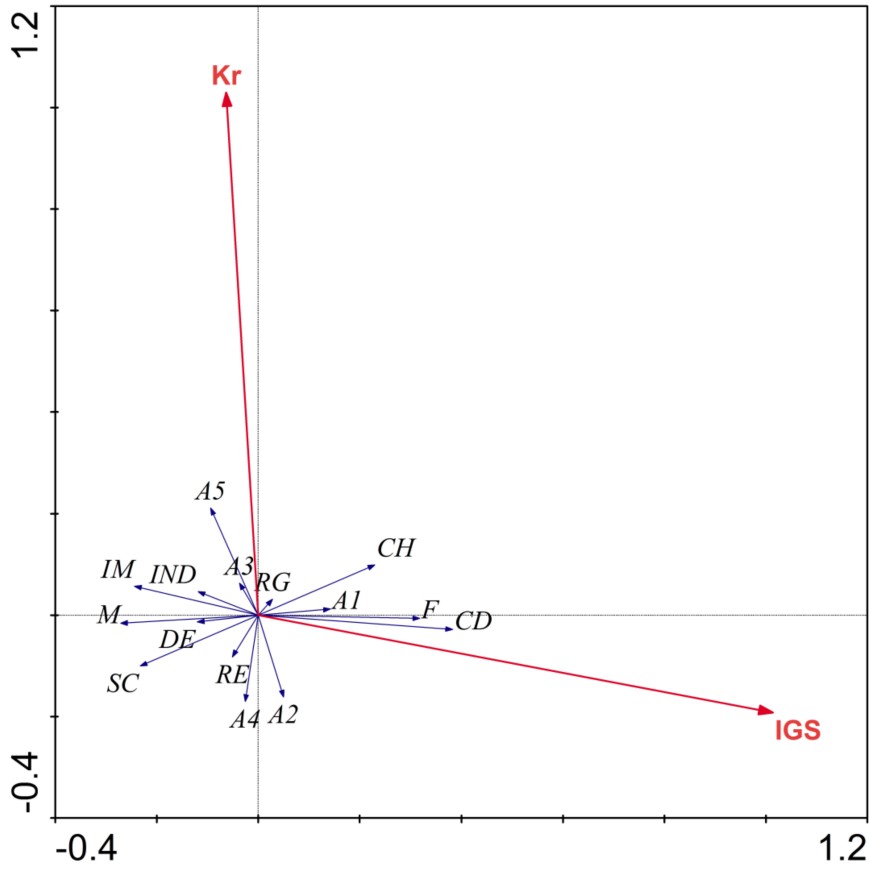

**Figure 7.** Ordination diagram (biplot) of the Redundancy Analysis (RDA) of reproductive indices and frequency of *Genyatremus luteus* specimens collected from April/16 to April/17 in Ajuruteua peninsula, Bragança, PA, Brazil. Seasonal periods (RP: rainy period, DP: dry period); maturation stages (IM: immature, DE: development, SC: spawning−capable, RG: regression, RE: regeneration, UND: undetermined), areas (A1: area 1, A2: area 2, A3: area 3, A4: area 4, A5: area 5).

## 4. Discussion

*G. luteus* (from 9.50 to 35.40 cm TL) were captured throughout this study, demonstrating this species uses this environment in all life stages at different periods and is estuarine-dependent [21,22]. Barletta-Bergan and Saint-Paul [8] in the same estuary found that juveniles of *G. luteus* were captured in both the dry period and rainy period; however, salinity and water temperature were crucial for the occurrence of the species. Like many species of fish, *G. luteus* is influenced by external factors, whether biotic (such as food availability) or abiotic, that trigger the physiological processes of reproduction, delimiting the optimal conditions for reproductive migration, spawning, and subsequent recruitment [23–25].

In this research, immature individuals of *G. luteus* occurred in almost all months, with greater records in the dry period. Compared to the study of Fernandes et al. [10], in a channel adjacent to the estuarine area of the Caeté River, the occurrence of immatures was directly influenced by lower precipitation, characterized by low turbidity and high salinity. This dynamic is a determinant for the entry and exit of *G. luteus* in high and low precipitation periods.

The frequent observations of immature individuals during the study period denotes that the species has a high frequency of cohorts entering the population, showing that the juveniles of *G. luteus* are estuarine-dependent during the crucial growth phase [8].

Noleto-Filho et al. [26] observed that the species has a synchronous ovary with spawning in two or more groups. The adaptation of the species to periodic spawning may result from the synchronization process between the optimization of gamete release and food availability for larvae and post-larvae [4]. This reproductive strategy of the species may be an alternative way to minimize interspecific competition between adult reproducers and their offspring, giving them a greater ability to adapt to different areas of the estuary and different periods of the year [27,28].

Spawning adults occurred more intensely in the rainy season. A similar result was found by Gómez et al. [29], between January and July in the Gulf of Paria in Venezuela. Marques et al. [30] observed, in a study with hormone induction via injection, that the reproductive success of the species is dependent on salinity, verifying that to activate motility, spermatozoa required polyhaline waters with a salinity of 30 ppt, and salinities of less than 20 ppt did not result in motile spermatozoa. Thus, the authors suggest that adult individuals live in coastal regions near the estuary, bay, or even in adjacent channels and tributaries with the influence of marine waters and, at the beginning of the rainy season, migrate to the lower estuary of the Caeté River until conditions are favorable for spawning.

The GSI and mean Kr values demonstrated the highest spawning intensities of the species occurred in the first six months of the year, with a peak in June, just after the period of heaviest rains in the region. The decreased values for these reproductive indices occurred mainly between April and September, demonstrating that the intensity of the 'internal response' to the 'environmental trigger' occurs soon after the heaviest rains. In this period, the Caeté River estuary is less influenced by freshwater, and with the decrease in rainfall, coastal marine waters enter in greater volumes, increasing salinities [31] and lowering tidal currents, wave strength, and height [32].

*G. luteus* females were larger in length and weight. Silva et al. [33] in a study of Pomadasys corvinaeformis, another Haemulidae, also verified sexual dimorphism, where males were smaller than females. In fish species, the individual reproductive capacity is proportional to the size and weight of the specimens [34], and these are related to fecundity [35].

In this study, the size of the first sexual maturity was estimated between 14.29 (males) to 15.13 cm (combined sexes), below the value of a 34.5 cm length (combined sexes) found by Gómez et al. [29] in the Gulf of Paria, Venezuela. According to that study, the species is subject to intense levels of exploitation by artisanal fisheries in that region, where 90% of the specimens were below the size of first maturity.

The difference between the sizes of adults recorded in this study and the study of Gómez et al. (2002) [29] was about 20 cm; according to Wootton (1990) [36], the first maturation can occur at different sizes depending on the environmental conditions of different environments and the intensity of fishing pressures [4]. Based on Fernandes et al. [9], fishing in the Caeté River estuary and coastal region means that 75.2% of landings come from small boats. These indications of the intensity of fishing activity in estuarine regions may compromise the growth of the species and culminate in a decrease in the average size of the first sexual maturity. Some environmental and anthropogenic drivers, including high rates of fishing mortality, cause impacts on size at maturity [37]. Moreover, when the effects of these drivers are intense and/or sustained over a long period of time, they could lead to evolutionary changes because of the selection of certain genotypes [38].

The ordination diagram showed that in the rainy season, the species behavior is associated with reproduction, where the mean values of GSI directly correspond to females with spawning capacities. The areas A1 and A2 confirmed the reproductive intensity in this period with the regressing-stage females. The highest mean values of Kr were associated mainly with immature and regressing individuals in areas A3 and A5. In these same areas (A3 and A5), in the dry period, the low GSI values were associated with the occurrence of immature, developing, and regenerating individuals, suggesting that these areas are used for growth, feeding, and the refuge of juveniles and adults.

The mean Kr values demonstrated that in the dry season, areas A2 and A4 are frequently used regardless of the life stage of the specimens. Additionally, as the species showed different behavior regarding the seasonal period, maturation stage, and functionality of the area in relation to reproduction, the aspects associated with population impacts generated by fishing need to be monitored.

From the data obtained here and in previous studies [9,10], and based on the life cycle of estuarine fishes [21], it is suggested that adults live in coastal regions, lower estuaries, and occasionally are found in upper estuaries during spawning season (Figure 8). The larvae are then carried by the tides to the interior of the estuary, where they grow to the juvenile stage (Figure 8). The pre-adults then migrate to areas outside the estuary and coastal regions to join the adult population and continue the life cycle of the species (Figure 8).

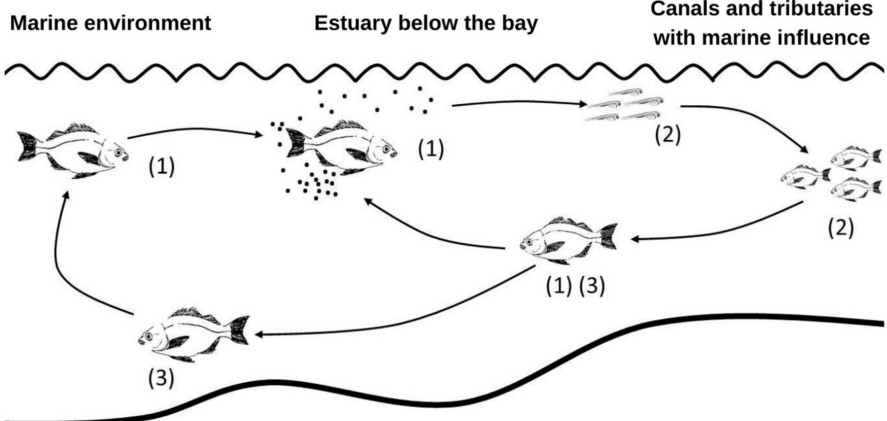

**Figure 8.** *Genyatremus luteus* life cycle. (1) Adults in the lower and bay and eggs dispersal; (2) larvae enter the estuary until juvenile stage; (3) pre-adults join the adult population, living in favorable areas until the spawning time of the species.

A management system needs to be implemented, even in Brazilian estuarine areas of artisanal fishing, once the effort is lower than it is offshore, as well as in industrial fishing areas where *G. luteus* is caught, too [39]. This research shows the first insights into a management policy for a species that requires different ecosystems to complete their life cycle, considering the lack of knowledge in relation to the Amazon reef [40].

**Author Contributions:** Conceptualization, S.F., L.P. and B.B.; methodology, S.F and B.B.; software, B.B.; validation, S.F, L.P. and B.B.; formal analysis, B.B.; research, S.F.; resources, S.F.; data curation, B.B.; draft preparation, S.F.; review and editing, L.P. and B.B.; visualization, B.B.; supervision, B.B.; project management, B.B.; funding acquisition, B.B. All authors have read and agreed to the published version of the manuscript.

**Funding:** This work was supported by the Conselho Nacional de desenvolvimento tecnológico CNPQ. The publication costs were obtained through the qualified publication program (PAPQ) of the Federal University of Pará.

**Institutional Review Board Statement:** This research is approved by Ethics committee of Federal University of Pará (CEUA), and the approval code is ID001884.

**Data Availability Statement:** The data sets analyzed while the study are not publicly available but can be made available to the corresponding author upon reasonable request.

**Acknowledgments:** Federal University of Pará and all those involved in the development and success of this study for their support.

**Conflicts of Interest:** The authors declare no conflict of interest.

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
