# Peer review of "Inferences about Fishing Impacts on the Reproductive Pattern of Torroto Grunt—Genyatremus luteus (BLOCH, 1790)—From the Amazon Continental Shelf"

_fishes, doi:10.3390/fishes8040181_

Round 1

Reviewer 1 Report

This is an interesting little paper which presents evidence of spawning season and maturation time. Overall the study is well presented. There are quite a few editorial issues, I have mentioned these in an attached annotated document. The English-language problem is minor, but there are still some grammatical fixes in areas that need to be made, hopefully i have identified all of these.

The biggest content issue is just the lack of a strong closing conclusive argument as to why this work is important. The authors have done the science, have done the analyses and presented the data in a mostly understandable format. The figures are all appropriate. They just need to tell us why in a stronger way why this work is important, and if possible, how it will benefit fishery managers in their area, or how the conclusions they have drawn could be useful to managers in a larger regional area. My recommendation is that this manuscript be subject to minor to moderate revision and then resubmitted to the journal for reconsideration.

Reviewer 2 Report

I found the article interesting but think using histologic maturity analyses would greatly improve confidence in the results. I am not familiar with the species studied but in many cases there are large differences detected in the age at maturity when the more sensitive method is used. But I do not think that is a fatal flaw and think the article should be published after a major rewrite - mainly focused on decreasing the wordiness and improving the English. In general some common errors were: 1) italicize scientific names throughout the manuscript; 2) use of commas instead of periods as decimal points in figures; 3) failure to capitalize the names of months; 4) overall wordiness. I think a major rewrite will be required before publication. I attached an edited version of the manuscript to help guide the authors in their revision.

Reviewer 3 Report

Comments to Authors:

Overall this paper examined the reproductive dynamics of Genyatremus luteus in an Amazonian estuary. The paper outlines important life-history characteristics such as timing of reproduction and length at first maturity. It also touches on potential impacts of fishing pressure on length at first maturity. Overall the authors have done a good job, however I have made several suggestions to improve the readability of the paper.

Major Comments:

1)  Abstract: The tile indicates fishing impacts on reproductive patterns will be a focus of the paper, yet in the abstract there is no mention of fishing impacts. Since the title gives the impression fishing impacts on the reproductive pattern of the species will be a focus, the abstract should make mention of the inferences about fishing impacts on the reproductive pattern that were made.

2)  The English used throughout the paper lacks clarity, I suggest the authors get a native English speaker to read the paper and edit it for grammar and clarity. I have made many suggestions throughout but use of a native English speaker to correct other instances of grammatical issues and lack of clarity would be very beneficial.

3)  Figure 6 is not uploaded in the paper, the graphs for figure 6 show the same graphs as figure 5. The authors must ensure the ordination biplot is uploaded as figure 6.

4) The discussion needs more detail and references to how fishing activity may reduce the L50. There needs to be some more directed discussion of this impact and the mechanism by which this happens if this is being included in the title. The authors do hypothesize how it happens but more directed discussion would be beneficial and help justify inclusion of “fishing pressure” in the title of the paper.

Line by Line Comments:

Title: “Inferences about fishing impacts on the reproductive pattern….” would sound better. Also I believe Torroto grunt should be two words not one.

Line 11: This sentence does not read well in English. Could say “Genyatremus luteus reproductive dynamics was investigated by examining gonadal first maturation length (L50) and reproductive period (gonadosomatic index - GSI and relative condition factor - Kr) in relation to the environmental factor rainfall.”

Line 12: Should be spelled “gonadosomatic index”, please change throughout the manuscript.

Line 20: The phrase “the grouped sex” does not sound correct, it could read “both sexes grouped together”

Line 11 and 38: The species name Genyatremus luteus should be italicized throughout the text. The common name Torroto grunt could also be given here?

Lines 38-40: This sentence does not make sense to me, consider rewording or clarifying.

Methods: Are the sampling, Maturity, Reproductive Cycle and Data analysis meant to be subsections, if so they should be given subsection heading.

Line 51: Sampling was conducted, not were conducted.

Line 50: Capitalize April here if you are capitalizing everywhere else.

Line 60-64: Was all this done in the lab? Were fish frozen and brought back to the lab to measure and extract gonads or was this done onsite after capture? Also were gonads weighed for the GSI calculation? These are important details that need to be stated clearly.

Line 71: Do you mean logistic regression instead of logistic graphical extrapolation?

Line 75: “Reproductive cycle” not “circle”.

Line 75-81: How was condition and GSI used to infer the reproductive cycle, this not clearly explained? Was a drop in GSI used to infer when reproduction occurs? The authors need to explicitly state, with references how condition and GSI were used to infer reproduction.

Line 99-100: overall females were larger than what? Males? Should also give the mean and sd of males for comparisons and could give results of a t-test that shows females were significantly larger than males.

Line 103: What statistical test did the p>0.05 result from, this could be stated in the brackets.

Line 128: Figure 3 should have a label on the y-axis

Line 140: The authors should be consistent with capitalizing months, I suggest capitalizing the names of months throughout the paper.

Line 146: Does not read well, could say “Dashed lines indicate no specimens were collected despite use of the same sampling effort.”

Line 147: Should read “ L50 was estimated to be 15.13 for females, 14,29 cm for males and 14.78 for both sexes grouped together.”

Line 148-149: This sentence does not make sense as is and needs to be reworded, I suggest “ The percent of adults captured according to the estimated L50s was 79.31 % for females, 68.57 % for males, and 71 % for both sexes grouped together.

Figure 5: The y-axis on figure 5 should read “Proportion of adults”, NOT “% Decimal adults”

Line 157: What is “it” referring to in this sentence, please clarify and be more explicit.

Figure 6: Currently figure 6 shows the same graphs as figure 5. Figure 6 is supposed to be a ordination biplot, the authors must ensure they upload the correct Figure 6 in a revision.

Line 169-170: This sentence does not read well, English editing needed here.

Line 207: What is meant by semester?

Round 2

Reviewer 2 Report

I think the manuscript is improved, the methods adequately described, the results clear and understandable and the discussion interesting. I also think the manuscript need to be strongly edited to improve the English but assume the journal editors will undertake that task but I am recommending accept after minor revision because of the writing style not being up to publication standards as yet. Thanks for completing the edits and changes I suggested originally.